# Homomorphism Autoencoder — Learning Group Structured Representations from Interactions

**Hamza Keurti**[1,2,3]    **Hsiao-Ru Pan**[1]    **Michel Besserve**[1]    **Benjamin Grewe**[2]    **Bernhard Schölkopf**[1]

[1]Max Planck Institute for Intelligent Systems, Tübingen, Germany
[2]ETH Zürich, Zürich, Switzerland,
[3]Max Planck ETH Center for Learning Systems

## Abstract

It is crucial for agents, both biological and artificial, to acquire world models that veridically represent the external world and how it is modified by the agent's own actions. We consider the case where such modifications can be modelled as transformations from a group of symmetries structuring the world state space. We use tools from representation learning and group theory to learn latent representations that account for both sensory information and the actions that alters it during interactions. We introduce the Homomorphism AutoEncoder (HAE), an autoencoder equipped with a learned group representation linearly acting on its latent space trained on 2-step transitions to implicitly enforce the group homomorphism property on the action representation. Compared to existing work, our approach makes fewer assumptions on the group representation and on which transformations the agent can sample from. We motivate our method theoretically, and demonstrate empirically that it can learn the correct representation of the groups and the topology of the environment. We also compare its performance in trajectory prediction with previous methods.

## 1 INTRODUCTION

An impressive feat of mammalian intelligence is the ability to learn effective internal models of the external world, allowing to infer key properties of the environment and to predict how it transforms under interventions. Sensorimotor interactions are likely crucial to learn such internal representations, enforcing them to evolve consistently with the external world. In the brain, information about performed actions taking the form of *efference copies*—copies of the motor signals sent to sensory regions of the brain—are exploited for processing sensory information and predicting future stimuli [Keller et al., 2012]. It is however unclear how sensory representations are intertwined with such representations of *interventions* performed by the agent, or even how the two are learned. Developmental psychologist Piaget [1964] postulates that representations of the world are learned from interaction starting from the first stage of development: the sensorimotor phase, and that these representations of the world are an understanding of how the world transforms under performed and imagined interventions.

The concept of group, as a mathematical structure, is pervasive in the description of the states and properties of the world. In physics, Noether's theorem shows the continuous symmetries of a system correspond to conserved quantities [Thompson and Cook, 1995]. Conversely, conserved quantities are associated with symmetries that only act on them while keeping everything else constant. Position in space is acted upon by the group of 3D translations, orientation by the group of 3D rotations and numerosity by the discrete group of linear translations. With the aim of representation learning being to recover these quantities from observed stimuli [Bengio et al., 2012], it is appropriate for the representation to transport the existing symmetries of the world states into the model. Such a symmetry-based representation satisfies an equivariance property between the true generating factors and the learned representations. To observe these symmetries, an agent can perform motor interactions to intervene on the world's state. These interactions also organize in a group which can be used to parametrize the symmetries of the real world.

While one can mathematically create infinitely many such representations, an agent with bounded computational abilities needs to choose one allowing efficient manipulation, interpretation and prediction of changes in its environment. A representational property compatible with this desideratum is disentanglement [Bengio et al., 2012, Kulkarni et al., 2015] which states the latent representation decomposes into parts reflecting the different interpretable properties of the environment that the agent can modify independently.

*Accepted for the 38*[th] *Conference on Uncertainty in Artificial Intelligence* (UAI 2022).

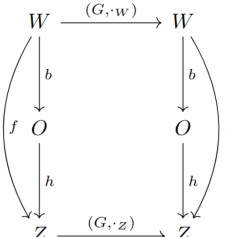

Figure 1: Commutative diagram for a symmetry-based representation.

While a group theoretic account of disentangled representations has been proposed by Higgins et al. [2018] without proposing a learning algorithm, few attempts have followed [Caselles-Dupré et al., 2019, Quessard et al., 2020], all assuming that the agent is able to separately intervene on independent factors of variation.

In this work, we propose the Homomorphism Autoencoder (HAE) framework to jointly learn a group representation of *transitions* between world states, as well as a symmetry-based disentangled representation of observations with minimal assumptions on the group or the actions the agent can sample from. We show theoretically and experimentally that the HAE learns the group structure of the set of transitions. In addition, the HAE learns to separate the pose of an object from the identity of acted-on objects, which can be identified as orbits of the non-transitive group action.

## 2 BACKGROUND

### 2.1 SYMMETRY-BASED DISENTANGLED REPRESENTATION LEARNING - SBDRL

Following the group theoretic formalism introduced by Higgins et al. [2018] (we refer to Appendix A for background on groups), we assume the set of observations $O \in R^{n_x \times n_y}$ is obtained from a set of world states $W$ through an unseen *generative process* $b : W \to O$. An *inference process* $h : O \to Z$ maps observations to their vector representations.

A group of symmetries $G$ structures the world states set by its action $\cdot_W : G \times W \to W$. $G$ is decomposed into a direct product of subgroups $G = G_1 \times ... \times G_n$. Here, each subgroup only transforms a specific latent property while keeping all others constant.

A representation is a symmetry-based representation if it verifies the commutative diagram in Figure 1. It then satisfies:

1. There is a (non-trivial) action of $G$ on $Z$:
$$\cdot_Z : G \times Z \to Z.$$

2. The composition $f = h \circ b : W \to Z$ is equivariant,

meaning that transformations of $W$ are reflected on $Z$. Formally, $f$ and the group action commute:

$$\forall g \in G, w \in W, \quad f(g \cdot_W w) = g \cdot_Z f(w).$$

The symmetry-based representation is disentangled with regard to the group decomposition $G = G_1 \times ... \times G_n$ if it satisfies this additional condition:

3. $Z$ can be written as a product of spaces $Z = Z_1 \times ... \times Z_n$ or as a direct sum of subspaces $Z = Z_1 \oplus ... \oplus Z_n$ such that each subgroup $G_i$ acts non trivially on $Z_i$ and acts trivially on $Z_j$ for $j \neq i$.

If we require the group action $\cdot_Z$ on $Z$ to be linear, then the existence of a group action on $Z$ is equivalent to the existence of a homomorphism $\rho : G \to GL(Z)$, called a group representation. The disentanglement condition is then expressed as follows:

3. (Linear) There exists a decomposition
$$Z = Z_1 \oplus ... \oplus Z_n$$
and a decomposition of the group representation
$$\rho = \rho_1 \oplus ... \oplus \rho_n$$
where each $\rho_i : G_i \to GL(Z_i)$ is a subrepresentation.

The action on $Z$ can then be written
$$g \cdot_Z z = \rho(g_1, ..., g_n)(z_1 \oplus ... \oplus z_n) \quad (1)$$
$$= \rho_1(g_1)z_1 \oplus ... \oplus \rho_n(g_n)z_n$$

for $g = (g_1, ..., g_n) \in G$ and $z = z_1 \oplus ... \oplus z_n \in Z$. Clearly each subgroup $G_i$ acts trivially on $Z_j$, $j \neq i$.

### 2.2 OBSERVING SYMMETRIES

Caselles-Dupré et al. [2019] proved that learning such symmetry-based disentangled representations requires observing the group elements that transition one state of the world into another (consequently one observation to the next). Similarly to past works [Quessard et al., 2020, Caselles-Dupré et al., 2019], we suggest using the interventions of an agent on its environment as a means to probe the symmetries of the environment. However, we do not assume that these interventions only act on one latent at a time. Instead, the agent intervenes through motor signals which are parametrized by the joints positions and it is not known a priori when an intervention only modifies one true latent while keeping others constant. The parametrization of the observed group elements is also expressed in the joints space instead of along the true latents. For a given group element, we will denote $\tilde{g}$, its parametrization in the latents space, and $g$ its parametrization in the joints space. And

we assume there exists a deterministic mapping between parametrizations $\varphi : \tilde{g} \mapsto g$. For instance, when a person moves a chalk along a blackboard, the chalk describes a 2D movement overparametrized by the rotation angles of the joints of the arm.

We propose to learn the inference process $h$ and a disentangled group representation $\rho$ using the HAE, described in section 3.2.

## 2.3 LIE GROUPS AND THE EXPONENTIAL MAP

We assume the group $G$ is a connected compact Lie group, the observed interventions are a discrete subset sampled from $G$. When a group $G$ is also a differentiable manifold, it is called a Lie Group. The tangent space to the group $G$ at the identity forms a Lie Algebra $\mathfrak{g}$: A vector space equipped with a bilinear product, the Lie bracket. We will leverage a convenient property of Lie Groups and their Lie Algebras that is the group can be studied through its tangent space. Indeed, the matrix exponential, called the exponential map in this setting, transports elements from the Lie Algebra to the Lie Group. Under certain assumptions, for instance if $G$ is connected compact, which we assume, the exponential map is surjective and therefore the whole group $G$ can be described from the tangent space at its identity. The exponential of an arbitrary matrix $A$ is given by the series $e^A = \sum_{k=0}^{\infty} \frac{1}{k} A^k$.

We construct the learnable group representation $\rho$ as the composition of the exponential map with an arbitrary mapping to the Lie Algebra.

$$\rho : G \xrightarrow{\phi} \mathfrak{g} \xrightarrow{exp} GL(Z)$$

We show how access to the group algebra can be leveraged in appendix C.2.3.

# 3 HAE ARCHITECTURE AND DERIVATION

## 3.1 TRANSITION DATASET

To illustrate our approach, we consider the dSprites dataset [Matthey et al., 2017], a labelled image dataset of 2D shapes (square, heart and ellipse) varied in scale, orientation and $x$ and $y$ positions, all of these factors of variations can be seen as the vector of latents $w$. We modify the dataset to observe transitions $(o_1, \tilde{g}_1, o_2, \tilde{g}_2, ..., \tilde{g}_{N-1}, o_n)$, where $\tilde{g}_i$'s correspond to differences of the latent vectors $w_{i+1}$ and $w_i$, they also parametrize the Lie Group of translations on the latents' manifold. As described in section 2.2, the agent is not provided the transitions $\tilde{g}$ but instead observes the performed actions $g = \varphi(\tilde{g})$. The agent is also not assumed to independently intervene on the different true latents [Caselles-Dupré et al., 2019, Quessard et al., 2020,

Painter et al.], instead interventions $\tilde{g}$ are sampled uniformly from a hypercube centered in identity. For most experiments, we limit ourselves to the cyclic translations group $G = C_x \times C_y$ of the $x$ and $y$ positions, cyclic in the sense that the vertical boundaries of the image are glued together and the horizontal boudaries are glued together, as such making the set of $(x, y)$ positions a torus. The agent observes $g = \varphi_{\pi/4}(\tilde{g})$ which rotates $\tilde{g}$ by $45 \deg$.

## 3.2 THE HOMOMORPHISM AUTOENCODER (HAE)

To jointly learn the latent representation $h$ of the observations and the group representation $\rho$, we introduce the 2-step HAE, described in Figure 2. An autoencoder equipped with a learnable mapping $\rho : G \rightarrow GL(Z)$ which maps an observed transition $g$ to an invertible matrix $\rho(g)$. The obtained matrix transforms encoding vectors of observations $z_i = h(o_i)$ to predict the encoding of future images. The latent prediction is evaluated on both the latent space through the latent prediction loss and on the image space through the reconstruction loss. $\gamma$ is a scalar hyperparameter.

$$\mathcal{L} = \mathcal{L}_{rec} + \gamma * \mathcal{L}_{pred}$$

We theoretically show in 3.3 how the proposed architecture leads to learning a symmetry-based representation $(\rho, h)$.

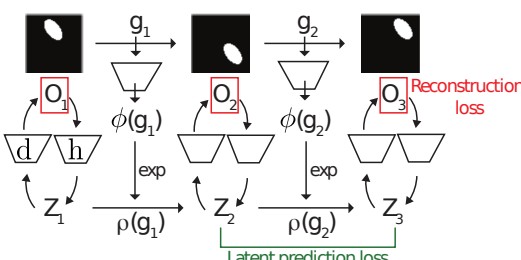

Figure 2: The Homomorphism Autoencoder consisting of $h$, $d$ and $\rho = exp \circ \phi$, relies on 2-step latent prediction to jointly learn the group representation $\rho$ and the observation representation $h$.

## 3.3 THE HAE LEARNS SYMMETRY-BASED REPRESENTATIONS

Previous attempts to design symmetry-based disentangled linear representations have put a lot of emphasis on the disentanglement property. However, it remains unclear how to learn a symmetry-based linear representation $(\rho, h)$ that verifies properties 1 and 2 in section 2.1, without enforcing strong assumptions on $\rho$ [Caselles-Dupré et al., 2019] or on the actions the agent can perform [Caselles-Dupré et al., 2019, Quessard et al., 2020].

In this section, we provide theoretical insights on learning symmetry-based representations and how the two-step HAE architecture achieves that with minimal assumptions.

We define the losses used throughout.

**The latent prediction loss** uses the group representation action to predict the evolution of stimuli encodings.

$$\mathcal{L}_{pred}^N(\rho, h) = \sum_t \sum_{j=1}^N ||h(o_{t+j}) - \prod_{i=0}^{j-1} \rho(g_{t+i})h(o_t)||_2^2$$

**The reconstruction loss** estimates the evolution of the observations from the predicted evolution of encodings. The reconstruction loss also evaluates the reconstruction of the initial observation like a standard autoencoder.

$$\mathcal{L}_{rec}^N(\rho, h) = \sum_t \sum_{j=0}^N ||o_{t+j} - d(\prod_{i=0}^{j-1} \rho(g_{t+i})h(o_t))||_2^2$$

The one step latent prediction loss is simply enforcing the commutative diagram in Figure 1. With the assumption that the group $G$ is a compact Lie Group, it admits a *faithful* group representation $\rho^*$ that we can assume is the one acting on the world states $W$. If we assume $\rho^*$ is given on $Z$, then minimizing $\mathcal{L}_{pred}^1(\rho^*, h)$ is enough to learn a symmetry-based representation.

**Proposition 1.** *Assume we observed the action of the group $G$ on each point of the observation space. Assume $h$ minimizes $\mathcal{L}_{pred}^1(\rho^*, h)$ then $h$ is a symmetry-based representation, meaning $h \circ b$ is equivariant.*

However, when $\rho^*$ is not known and a group representation $\rho$ of $G$ needs to be learned over a space of arbitrary mappings, minimizing $\mathcal{L}_{pred}^1(\rho, h)$ can lead to the trivial representation.

**Proposition 2.** *The trivial group representation $\rho = I$ (that always maps to the identity matrix) combined with a constant $h$ is a zero of the prediction loss $\mathcal{L}_{pred}^1(\rho, h)$.*

The reconstruction loss of the initial observation helps avoid the representation collapse into a trivial solution by ensuring $h$ is not constant for a given fixed group representation $\rho^0$. Although we found using $\mathcal{L}_{rec}^2(\rho, h)$ works better than $\mathcal{L}_{rec}^0(\rho, h)$ when jointly learning $(\rho, h)$.

**Proposition 3.** *Assume the data samples at least once all points of the observation space. If $h$ minimizes $\mathcal{L}_{rec}^N(\rho^0, h)$ then $h$ is injective.*

We now present the main theoretical result of the paper: The HAE through enforcing the 2-step latent prediction loss and the observations reconstruction (enforcing $h$ is injective) learns a symmetry-based representation.

**Proposition 4.** *Assume $(\rho, h)$ minimizes $\mathcal{L}_{pred}^2(\rho, h)$ and $h$ is injective, then $\rho$ is a non-trivial group representation and $(\rho, h)$ is a symmetry-based representation.*

## 3.4 DISENTANGLEMENT

As expressed in section 2.1 the disentanglement condition for a linear action on $Z$ defined through its group representation $\rho$, is a decomposition of both the representation space $Z = \bigoplus_1^n Z_i$ and the group representation $\rho = \bigoplus_{i=1}^n \rho_i$. Where the subgroup representations $\rho_i$ are representations of the subgroups $G_i$ on the subspaces $Z_i$.

Following that the group G is decomposed in the true latent's parametrization, the observed group representation is disentangled with regard to the group decomposition if in matrix form, the group representation of any group element $g = \varphi(\tilde{g})$ is a block-diagonal matrix of the subgroups representations:

$$\rho\big(g = \varphi(\tilde{g}^1, ..., \tilde{g}^n)\big) = \begin{pmatrix} \rho_1(\tilde{g}^1) & 0 & \cdots & 0 \\ 0 & \rho_2(\tilde{g}^2) & \ddots & \vdots \\ \vdots & \ddots & \ddots & 0 \\ 0 & \cdots & 0 & \rho_n(\tilde{g}^n) \end{pmatrix}$$
(2)

We can therefore constrain our trainable group representation in the space of matrices of the block diagonal form given in equation 2. This requires prior knowledge of:

- The number of groups in the decomposition.

- The dimension of each subgroup representation $dim(Z_i)$.

We start by assuming knowledge of this information, however we hope that in future works, one would be able to search for the best decomposition. However, we do not assume prior knowledge of the decomposition. Meaning that given an observed transition $g$, we do not have access to the decomposition $\varphi^{-1}(g) = \tilde{g} = (\tilde{g}^1, ..., \tilde{g}^n)$ along the group decomposition $G = G_1 \times ... \times G_n$. Which means we are learning representations of the form given in equation 3.

$$\rho(g) = \begin{pmatrix} \rho_1(g) & 0 & \cdots & 0 \\ 0 & \rho_2(g) & \ddots & \vdots \\ \vdots & \ddots & \ddots & 0 \\ 0 & \cdots & 0 & \rho_n(g) \end{pmatrix}$$
(3)

While we do not prove that this block diagonal constraint leads to disentanglement, we show through experiments that HAE learns a symmetry-based representation $(\rho, h)$ that is disentangled with regard to the true factors and takes the form in equation 2.

# 4 EXPERIMENTS

## 4.1 LEARNING A 2-TORUS REPRESENTATION MANIFOLD

We consider a subset of the dsprites dataset [Matthey et al., 2017] where a fixed scale and orientation ellipse is acted on by the group of 2D cyclic translations $G = C_x \times C_y$. The corresponding transition dataset contains tuples $(o_1, g_1, o_2, ..., g_{n-1}, o_n)$, where the observations $o_i$ are $64 \times 64$ pixels and the transitions are given by $g_i = \varphi_{\pi/4}(\tilde{g}_i)$ where $\tilde{g}_i = (\tilde{g}_i^x, \tilde{g}_i^y)$ parametrizes the displacement along $x$ and $y$, and $\varphi_{\pi/4}$ is the rotation of the 2D plane by $45$ deg. We train the 2-step HAE described in section 3.2 with a 4D latent space and with the group morphism $\rho = \exp \circ \phi$ where $\phi$ maps to a 2-blocks diagonal matrix each of dimension $2 \times 2$. Architecture and hyperparameters for training are specified in the appendix.

**Learned data representation** We visualize the learned 4-dimensional encodings of the whole dataset through 2-dimensional random matrix projections, and selected few matrices showing the most discernable projections (Figure 3). The learned manifold corresponds to the expected latent space topology $S^1 \times S^1$.

**Learned Group Representation $\rho$** We then evaluate the learned matrices for the identity $id = (0, 0)$, the generating elements of each subgroup $\tilde{1}_x = (1, 0), \tilde{1}_y = (0, 1)$ and their inverses $\tilde{g}_x^{-1} = (-1, 0), \tilde{g}_y^{-1} = (0, -1)$, Figure 4. Note that the corresponding observed actions $g$ presented to the group representation $\rho$ are in the same order: $(0, 0), \frac{\sqrt{2}}{2}(1, 1), \frac{\sqrt{2}}{2}(-1, 1), \frac{\sqrt{2}}{2}(-1, -1), \frac{\sqrt{2}}{2}(1, -1)$.

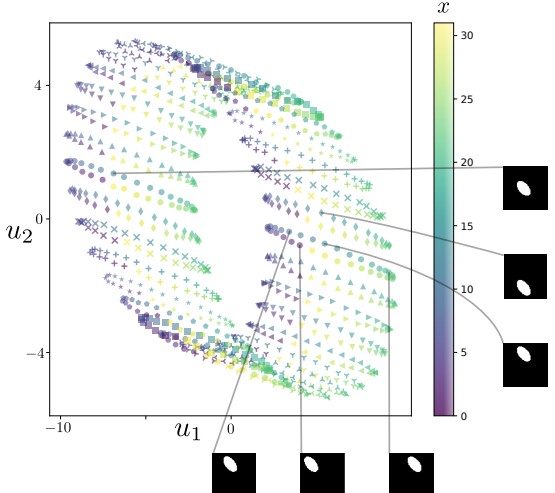

Figure 3: Random 2D projection of the 4D HAE latent encodings of the translated ellipse dataset. Color indicates $y$ position of the ellipse, while markers indicate $x$ position.

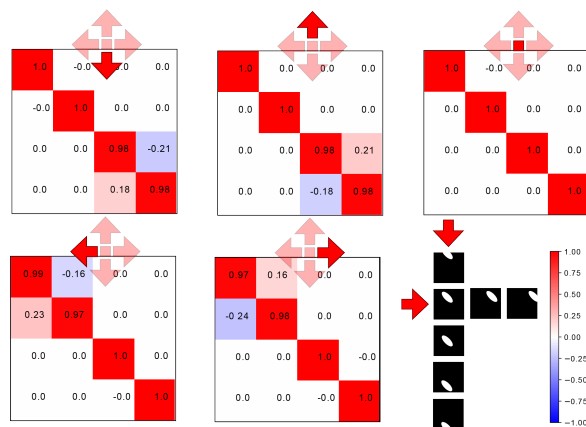

Figure 4: Evaluation of the learned and disentangled group representation $\rho$ for the identity element (upper right) and generative transitions (middle and left) yields block rotation matrices.

The matrices obtained, for example, actions in Figure 4, show that $\rho(0) = I_4$ and that representations for elements belonging to subgroups of the decomposition $G = C_x \times C_y$ follow the disentangled group representation predicted in subsection 3.4 where actions on the same subgroup have representations that act on the same subspace while fixing the other subspace. In addition, the blocks correspond to 2D rotation matrices which are of the form:

$$R(\theta) = \begin{pmatrix} \cos(\theta) & -\sin(\theta) \\ \sin(\theta) & \cos(\theta) \end{pmatrix}$$

The angle of rotation of an elementary step corresponds to the number of equally spaced true latent values for each subgroup: $2\pi/32$, with $\cos(2\pi/32) \approx 0.981$ and $\sin(2\pi/32) \approx 0.195$. Additional information on this setup is available in the Appendix C.2.

## 4.2 ROLLOUT PREDICTION

One important application of learning structured representation is to predict how the observations would change given sequences of actions. We compare HAE to two other approaches of modeling the dynamics in the latent space: (1) *Unstructured*: $z_{t+1} = h(z_t, g_t)$, where $h$ is a learnable function. Similar approaches have been widely adapted in recent model-based deep RL methods [Ha and Schmidhuber, 2018, Schrittwieser et al., 2020]. (2) *Rotations*: $z_{t+1} = R_g z_t$, where $R_g = \prod_{i,j} G(i, j, \theta_{ij,g})$ and $G(i, j, \theta_{ij,g})$ are the Givens rotation matrices. This approach was proposed by Quessard et al. [2020] and was shown to be capable of learning symmetry-based representations when the actions are sampled along the true latents.

We evaluate the methods in an offline setting, where we train each method on a given set of 2-step trajectories and test their generalization ability on a hold-out set of 128-step tra-

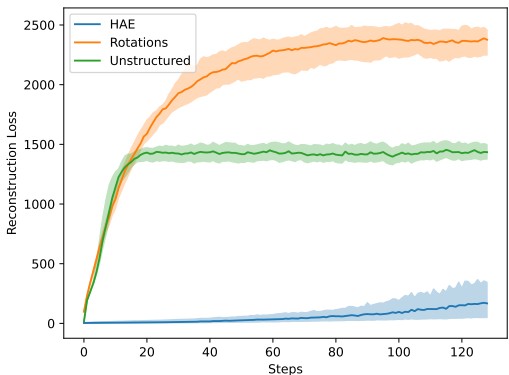

Figure 5: Step-wise reconstruction loss on the test dataset. Lines and shadings represent median and interquartile range over 50 random seeds.

jectories. See Appendix C.3 for details on the setup. Figure 5 shows the reconstruction loss for each method on the test trajectories. Our result suggests that when the actions sampled are not disentangled (i.e., each action may involve changes in multiple generating factors), the Rotations method may perform worse than the Unstructured method while HAE can outperform them significantly.

### 4.3 UNSUPERVISED IDENTITY SEPARATION FROM INTERVENTION

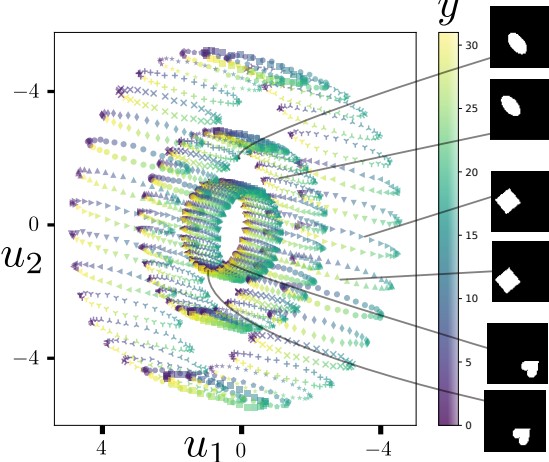

Figure 6: $5D$ representation space of a trained homomorphism autoencoder with two $2D$ group subrepresentations and a $1D$ representation space acted on trivially. The representation vectors for the whole dataset are projected to $2D$ via Random Projection. Colors indicate $y$ position, markers indicate $x$ position and each torus corresponds to a separate shape.

We consider the subset of the dSprites dataset consisting of three shapes (heart, square and ellipse) acted on by the previous group of 2D cyclic translations $G = C_x \times C_y$.

The action of the group is not *transitive*, as it describes a separate *orbit* for each shape (see appendix A). Indeed, no intervention changes the shape. In this experiment, we augment the representation space by one dimension to account for the shape property, and the group representation is fixed to act trivially on it.

We learn 5-dimensional encodings of the observations $2 \times 2D$ subspaces acted on by the subrepresentations of the group $G$ and 1 representation unit trivially acted on.

Our results in Figure 6 show that the model not only learns the representation of the cyclic translation group action shared among shapes, but also learns to separate the representation of observations by shape along the last $G$-invariant representation unit, giving rise to three identical manifolds. This is reminiscent of the two-streams hypothesis of visual processing [Goodale and Milner, 1992], the "What" pathway processes information related to object identity, while the "Where" pathway processes information related to the object pose, or the required motor action for manipulation.

A more thorough analysis of the experiment can be found in Appendix C.4.

## 5 DISCUSSION

We provide theoretical and experimental justification that the HAE allows an agent to extract geometric structure of its external world while learning a similarly structured, low dimensional internal manifold. In contrast to earlier works, our only prior assumption is the number of subgroups in the group decomposition, as well as the dimensionsionality to represent each. When using a set of cyclic actions, we find that the HAE maps the geometric action structure into the agent's latent space, in the sense that disentangled geometric latent variables representing the motion factors as well as non-geometric variables representing different objects or shapes emerge. In particular, the emergence of invariant object representations provides a new angle for those seeking to learn rich and behavior-relevant representations of objects without the need for labels. Finally, it is interesting to note that similar cyclic embeddings have been reported in neuroscience, for example, in the hippocampus of mice where toroid-like embeddings encode the animals' head-direction [Chaudhuri et al., 2019]. In fact, our cyclic shape translation task restricted to a single dimension could be viewed as an agent horizontally rotating its head while observing its environment. One limitation of our theoretical analysis is the deterministic nature of our model, the intrinsically non-linear nature of the problem would make the theoretical analysis more challenging in the stochastic setting. On the experimental side, we solely assessed the extraction and internal representation of geometric action structure, we leave it to future work to test if the same principled HAE approach generalizes to learn other, non-cyclic, facets of the structure of the external world.

## Acknowledgements

Hamza Keurti is grateful to CLS for generous funding support. The authors thank the International Max Planck Research School for Intelligent Systems (IMPRS-IS) for supporting Hsiao-Ru Pan. Further this work was supported by the Swiss National Science Foundation (B.F.G. CRSII5-173721 and 315230_189251), ETH project funding (B.F.G. ETH-20 19-01), the Human Frontiers Science Program (RGY0072/2019).

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

# A    BACKGROUND ON GROUP THEORY

In this section, we provide an overview of group theory concepts exploited in this work.

**Definition A.1** (Group). *A set $G$ is a group if it is equipped with a binary operation $\cdot : G \times G \to G$ and if the group axioms are satisfied*

1. *Associativity:* $\forall a, b, c \in G, (a \cdot b) \cdot c = a \cdot (b \cdot c)$

2. *Identity: There exists $e \in G$ such that $\forall a \in G, a \cdot e = e \cdot a = a$.*

3. *Inverse:* $\forall a \in G$, there exists $b \in G$ such that $a \cdot b = b \cdot a = e$. This inverse is denoted $a^{-1}$.

We are often interested in sets of transformations, which respect a group structure, but are applied to objects that are not necessarily group elements. This can be studied through group actions, which describe how groups *act* on other mathematical entities.

**Definition A.2** (Group Action). *Given a group $G$ and a set $X$, a group action is a function $\cdot_X : G \times X \to X$ such that the following conditions are satisfied.*

1. *Identity: If $e \in G$ is the identity element, then $e \cdot_X x = x, \forall x \in X$.*

2. *Compatibility: $\forall g, h \in G$ and $\forall x \in X$, $g \cdot_X (h \cdot_X x) = ((g \cdot h) \cdot_X x)$*

We restrict ourselves to learning representations that are structured linearly by the group, where the action of each group element on our representation space is described by an invertible matrix, that we identify with the set of linear invertible transformations $GL(V)$ of a finite-dimensional vector space $V$. This mapping is called a group representation. Actions of this type have been studied extensively in representation theory.

**Definition A.3** (Group Representation). *Let $G$ be a group and $V$ a vector space. A representation is a function $\rho : G \to GL(V)$ such that $\forall g, h \in G$, one has $\rho(g)\rho(h) = \rho(g \cdot h)$.*

Note that such definition is not restricted to finite dimensional vector spaces, however we will limit our study to this case, such that representations are appropriately described by mappings from $G$ to a space of square matrices.

**Definition A.4** (Lie Group). *A Lie Group $G$ is a nonempty set satisfying the following conditions:*

- *$G$ is a group.*

- *$G$ is a smooth manifold.*

- *The group operation $\cdot : G \times G \to G$ and the inverse map $.^{-1} : G \to G$ are smooth.*

We limit ourselves to the study of linear Lie Groups, Lie groups that are matrix groups. The tangent space to a Lie Group at the identity forms a Lie Algebra. A Lie Algebra $\mathfrak{g}$ is a vector space equipped with a bilinear map $[.,.] : \mathfrak{g} \times \mathfrak{g} \to \mathfrak{g}$ called the Lie Bracket. We will not introduce the Lie Bracket as we do not make use of it. The Lie Algebra somehow describes most of everything happening in its Lie Group. This connection is established through the exponential map.

**Definition A.5** (Exponential Map). *The exponential map $exp : \mathfrak{g} \to G$ is defined for matrix Lie Groups by the series:*

$$e^A = \sum_{k=0}^{\infty} \frac{1}{k} A^k. \quad \forall A \in \mathfrak{g}$$

The exponential map is not always surjective. However if we only consider groups that are connected and compact, the exponential is surjective, which justifies our parametrization of the group representation through:

$$\rho : G \xrightarrow{\phi} \mathfrak{g} = M_n(\mathbb{R}) \xrightarrow{exp} GL_n(\mathbb{R})$$

Where $\phi$ is a trainable arbitrary mapping.

**Group action types** The effect of a group action on a base space $X$ varies according to the properties of the homomorphism defined by the group action

$$\tau : G \to Sym(X)$$
$$g \mapsto g \cdot_X \square$$

We introduce two types of actions:

**Definition A.6** (Transitive Group Action). *The action of $G$ on $X$ is* transitive *if $X$ forms a single orbit.*

*in other words, $\forall x, y \in X, \exists g \in G; g \cdot x = y$.*

**Definition A.7** (Faithful Group Action). *The action of $G$ on $X$ is* faithful *if the homomorphism $G \to Sym(X)$ corresponding to the action is bijective (an isomorphism).*

*In that case, $\forall g_1 \neq g_2 \in G, \exists x \in X; \quad g_1 \cdot x \neq g_2 \cdot x$.*

We also define the *orbits* by a group action:

**Definition A.8** (Orbit by a Group Action). *The orbit of an element $x \in X$ by the action $\cdot_X$ of a group $G$ is the set*

$$G \cdot_X x = \{ g \cdot_X x : g \in G \}$$

When the action of $G$ is transitive on $X$, then $X$ is the single orbit by the action of $G$:

$$\forall x \in X, G \cdot_X x = X$$

Such is the case for our experiments using a single shape. We also explore the case where the action is not transitive in the multi shape experiment visualized in Figure 6.

# B THEORETICAL RESULTS

We first prove the main theoretical result of the paper, then proceed to prove other propositions.

**Proposition 4.** *Assume $(\rho, h)$ minimizes $\mathcal{L}^2_{pred}(\rho, h)$ and $h$ is injective, then $\rho$ is a non-trivial group representation and $(\rho, h)$ is a symmetry-based representation.*

*Proof.* Given that the group $G$ is supposed compact, it admits a group representation by the *Peter-Weyl theorem*. We can therefore assume that the true state space $W$ is acted on linearly by $G$ through its representation $\rho^*$. As such the inverse of the generating process $b^{-1}$ and $\rho^*$ verify $\mathcal{L}^2_{pred}(\rho^*, b^{-1}) = 0$.

Assume $h$ is injective, guaranteed by a minimization of the 0-step reconstruction loss.

Assume $(\rho, h)$ minimizes $\mathcal{L}^2_{pred}(\rho, h)$ therefore $\mathcal{L}^2_{pred}(\rho, h) = 0$.

Then for all observed 2-step transitions $(o_t, g_t, o_{t+1}, g_{t+1}, o_{t+2})$ — note that observed transitions $(o_t, g_t, o_{t+1})$ correspond to an action on the true world states $w_{t+1} = g_t \cdot_W w_t$ — $(\rho, h)$ verifies:

$$\rho(g_t)h(o_t) = h(o_{t+1})$$

and

$$\rho(g_{t+1})\rho(g_t)h(o_t) = h(o_{t+2})$$

Let us prove $\rho$ is a group representation, meaning it verifies $\rho(g_2 g_1) = \rho(g_2)\rho(g_1), \forall g_1, g_2 \in G$.

Let $g_1, g_2, g_3 \in G$ such that $g_3 = g_2 g_1$.

Let $w_1, w_2, w_3$ which verify $w_2 = g_1 \cdot_W w_1$, $w_3 = g_2 \cdot_W w_2$, $w_3 = g_3 \cdot_W w_1$. Therefore, the associated transitions $(o_1, g_1, o_2)$, $(o_2, g_2, o_3)$ and $(o_1, g_3, o_3)$ can be observed.

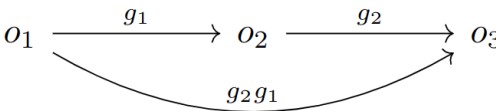

Assume, we observe the 2-step transitions $(o_t, g_t, o_{t+1}, g_{t+1}, o_{t+2})$ and $(o_{t'}, g_{t'}, o_{t'+1}, g_{t'+1}, o_{t'+2})$ such that:

$$\begin{cases} o_t = o_{t'} = o_1 \\ o_{t+1} = o_2 \\ o_{t+2} = o_{t'+1} = o_3 \end{cases}$$

and

$$\begin{cases} g_t = g_1 \\ g_{t+1} = g_2 \\ g_{t'+1} = g_3 = g_2 g_1 \end{cases}$$

$\mathcal{L}^2_{pred} = 0$ gives for the transition at $t$: $\rho(g_{t+1})\rho(g_t)h(o_t) = h(o_{t+2})$ therefore $\rho(g_2)\rho(g_1)h(o_1) = h(o_3)$.

And for the transition at $t'$: $\rho(g_{t'})h(o_{t'}) = h(o_{t'+1})$ therefore $\rho(g_3)h(o_1) = h(o_3)$

Therefore we have $\rho(g_2)\rho(g_1)h(o_1) = \rho(g_3)h(o_1)$ or $\rho(g_2)\rho(g_1)h(o_1) = \rho(g_2 g_1)h(o_1)$

With this equality verified over the set $O_{tr}$ of first observations $o_1$ in the training set — meaning we observe two successions of the action of group elements $(g_1, g_2)$ and $g_3$ for different values of the starting observation $o_1$ — by assuming $h(O) \subseteq span(h(O_{tr}))$,

we get that $\rho(g_2)\rho(g_1) = \rho(g_2 g_1)$ over $h(O)$ (equality of linear mappings over vectors that span a vector subspace).

Therefore the subrepresentation of $\rho$ over $span(h(O))$ is a group representation of $G$.

The injectivity assumption ensure $h(O)$ does not collapse to a single element.

Let us show $h$ is a symmetry based representation.

We have for every observed transition $(o_t, g_t, o_{t+1})$:

$$h(o_{t+1}) = \rho(g_t)h(o_t)$$

by the generative model assumptions

$$o_{t+1} = b(w_{t+1}) = b(g_t \cdot_W w_t)$$

Also $o_t = b(w_t)$, finally

$$h \circ b(g_t \cdot_W w_t) = \rho(g_t)h \circ b(w_t)$$

$\square$

**Proposition 1.** *Assume we observed the action of the group $G$ on each point of the observation space. Assume $h$ minimizes $\mathcal{L}^1_{pred}(\rho^*, h)$ then $h$ is a symmetry-based representation, meaning $h \circ b$ is equivariant.*

*Proof.* Given that the group $G$ is supposed compact, it admits a group representation by the *Peter-Weyl theorem*. We can therefore assume that the true state space $W$ is acted on linearly by $G$ through its representation $\rho^*$. As such the inverse of the generating process $b^{-1}$ and $\rho^*$ verify $\mathcal{L}^2_{pred}(\rho^*, b^{-1}) = 0$.

As a consequence, $h$ also achieves zero loss such that

$$\sum_t \sum_{j=1}^{N} ||h(o_{t+j}) - \prod_{i=0}^{j-1} \rho^*(g_{t+i})h(o_t)||_2^2$$

such that for all $(j, t)$

$$h(o_{t+j}) = \prod_{i=0}^{j-1} \rho^*(g_{t+i})h(o_t)$$

In particular for $j = 1$

$$h(o_{t+1}) = \rho^*(g_t)h(o_t)$$

by the generative model assumptions

$$o_{t+1} = b(w_{t+1}) = b(\rho^*(g_t)w_t)$$

Also $o_t = b(w_t)$ such that

$$h \circ b(\rho^*(g_t)w_t) = \rho^*(g_t)h \circ b(w_t)$$

$\square$

**Proposition 3.** *Assume the data samples at least once all points of the observation space. If $h$ minimizes $\mathcal{L}_{rec}^N(\rho^0, h)$ then $h$ is injective.*

*Proof.* Assume the encoder $h$ and the decoder $d$ minimize the 0-step reconstruction.

$$\forall o, o' \in O, \text{ such that } o \neq o'.$$

$$d(h(o)) = o \text{ and } d(h(o')) = o'$$

Therefore

$$d(h(o)) \neq d(h(o')).$$

Finally

$$h(o) \neq h(o')$$

Therefore $h$ is injective.

$\square$

**Proposition 5.** *Assume $h = h^*$ disentangled, equivariantly maps observations in $O$ to $Z$ such that there exists a non-trivial disentangled linear action $\rho^*$ of $G$ on $Z$ $\rho^*$ according to the group decomposition $G = G_1 \times ... \times G_n$.*

*Assume $\rho$ of the form in equation 3 minimizes $\mathcal{L}_{pred}^2(\rho, h^*)$ then $\rho$ is a group representation and $\rho$ defines the same action as $\rho^*$ over the subspace spanned by $h^*(O)$.*

*Proof.* We assume the existence of $(h^*, \rho^*)$ such that $\rho^*$ is a disentangled representation of $G$ with regard to the decomposition $G = G_1 \times ... \times G_n$ and $h^*$ is an equivariant representation with regard to $\rho^*$.

As such, $\rho^*$ verifies:

$$\rho^* = \rho_1^* \oplus ... \oplus \rho_n^*$$

such that

$$\forall i, \forall g = (g_1, ..., g_n) \in G_1 \times ... \times G_n; \rho_i^*(g) = \rho_i^*(g_i)$$

$h^*$ verifies

$$h^* \circ b(g \cdot_W w_t) = \rho(g) h^* \circ b(w_t)$$

equivalently, in terms of an observed transition $(o_t, g, o_{t+1})$:

$$h^*(o_{t+1}) = \rho^*(g) h^*(o_t)$$

We assume $h^*$ is given but not $\rho^*$. If $\rho = \rho_1 \oplus ... \oplus \rho_n$ satisfies $\mathcal{L}_{pred}^2(\rho, h^*) = 0$, then for a given $g \in G$:

$$h^*(o_{t+1}) = \rho(g) h^*(o_t)$$

Then:

$$\rho^*(g) h^*(o_t) = \rho(g) h^*(o_t)$$

is verified for all observations $o_t \in O$. Which leads to the equality of the matrices over the subspace spanned by $h(O)$.

Note that for each $g$, it is enough to observe the transitions $(o_t, g, o_{t+1})$ such that $o_t \in O_S$ such that $h(O)$ is in the span of $h(O_S)$.

$\square$

**Proposition 6.** *Assume $\rho = \rho^*$ a non-trivial disentangled group representation of $G$ on $Z$.*

*Assume $h$ minimizes $\mathcal{L}_{pred}^2(\rho^*, h)$ then $h$ is a symmetry based disentangled representation with regard to the disentangled group representation $\rho^*$.*

*Proof.* Given that the group $G$ is supposed compact, it admits a group representation by the *Peter-Weyl theorem*. We can therefore assume that the true state space $W$ is acted on linearly by $G$ through its representation $\rho^*$. As such the inverse of the generating process $b^{-1}$ and $\rho^*$ verify $\mathcal{L}_{pred}^2(\rho^*, b^{-1}) = 0$.

Assume $h$ minimizes $\mathcal{L}_{pred}^2$ then it achieves zero loss such that

$$\sum_t \sum_{j=1}^N ||h(o_{t+j}) - \prod_{i=0}^{j-1} \rho^*(g_{t+i}) h(o_t)||_2^2$$

such that for all $(j, t)$

$$h(o_{t+j}) = \prod_{i=0}^{j-1} \rho^*(g_{t+i}) h(o_t)$$

In particular for $j = 1$

$$h(o_{t+1}) = \rho^*(g_t) h(o_t)$$

by the generative model assumptions

$$o_{t+1} = b(w_{t+1}) = b(g_t \cdot_W w_t)$$

Also $o_t = b(w_t)$, finally

$$h \circ b(g_t \cdot_W w_t) = \rho^*(g_t) h \circ b(w_t)$$

$\square$

## C  EXPERIMENTS

### C.1  DATA

We use a subset of the dSprites dataset consisting only of the ellipse at a fixed scale and orientation with varying $x$ and $y$ positions, there are 32 equally spaced positions for each. We consider that the ellipse is acted on by the group $G = G_x \times G_y$ of cyclic translations, where the sprite warps to the opposite extremity when it reaches an extremal position. For each image $o_1$ we sample group elements $\tilde{g}_1 = (\tilde{g}_x, \tilde{g}_y)$ uniformly from a square around identity spanning the range $[\![-10, 10]\!]$. We assume the agent observes $g_1 = \varphi_{45}(\tilde{g}_1) = \frac{\sqrt{2}}{2}(\tilde{g}_x - \tilde{g}_y, \tilde{g}_x + \tilde{g}_y)$. We obtain the first transition $(o_1, g_1, o_2)$, we do the same for $o_2$ to get 2-step transitions $(o_1, g_1, o_2, g_2, o_3)$.

## C.2 LEARNING A DISENTANGLED REPRESENTATION

### C.2.1 Hyperparameters

**Model architecture** We use a symmetrical architecture for the encoder and decoder, which we summarize in Table 1. The network was trained on the combined loss:

$$\mathcal{L} = \mathcal{L}_{rec}^2(\rho, h) + \gamma \mathcal{L}_{pred}^2(\rho, h)$$

Where we use the Binary Cross Entropy loss for the reconstruction term instead of the Mean Squared Error as it is better behaved during training.

Table 1: Network architecture.

| Parameter | Value |
|---|---|
| Conv. Channels | [64, 64, 64, 64] |
| Kernel Sizes | [6, 4, 4, 4] |
| Strides | [2, 2, 1, 1] |
| Linear Layer Size | 1024 |
| Activation | ReLU |
| Latent space | 4 |
| $\gamma$ | 400 |
| Group representation dimensions | [2,2] |

**Training hyperparameters** We trained the network using the hyperparameters summarized in Table 2.

Table 2: Training hyperparameters.

| Parameter | Value |
|---|---|
| Optimizer | Adam |
| Learning rate | 0.001 |
| Number of training sequences | 10000 |
| Batch size | 500 |
| Epochs | 101 |

### C.2.2 Visualization

We obtained Figure 3 by projecting the 4D representation vector for each image in the dataset on a random 2D plane through Random Matrix Projection. We chose the projection with the most explainable visualization.

### C.2.3 Latent Traversal

We show how learning a mapping to the group algebra can be leveraged to navigate the group and the data manifold. We remind that $\rho = \exp \circ \phi$, where $\phi$ maps to the algebra $\mathfrak{g}$ of the group $G$, and $\exp$ is the matrix exponential which gives a connection between the algebra and the lie group.

The mapping $\phi = \phi_1 \oplus \phi_2$ and the group representation $\rho = \rho_1 \oplus \rho_2$ are constrained on the space of block diagonal matrices of the form $M = M_1 \oplus M_2$ each of dimension $2 \times 2$. However since each block is made of elements from a $1D$ subgroup of $GL_2(\mathbb{R})$: $SO(2)$, its algebra is the $1D$ subalgebra of $M_2(\mathbb{R})$ of skew-symmetric matrices. We find this subspace by performing a PCA over each of the sets $\{\phi_1(g_t)\}_t$ and $\{\phi_2(g_t)\}_t$ for a batch $\{g_t\}_t$ of observed transitions. The first component for each block $E_1$ and $E_2$ corresponds to the only base vector for that subalgebra. We find:

$$E_1 \oplus 0_{2,2} = \begin{bmatrix} -0.04 & -0.65 & 0 & 0 \\ 0.76 & -0.04 & 0 & 0 \\ 0 & 0 & 0 & 0 \\ 0 & 0 & 0 & 0 \end{bmatrix}$$

$$0_{2,2} \oplus E_2 \approx \begin{bmatrix} 0 & 0 & 0 & 0 \\ 0 & 0 & 0 & 0 \\ 0 & 0 & 0.01 & -0.65 \\ 0 & 0 & 0.76 & 0.01 \end{bmatrix}$$

We obtain the figure 7 by linearly traversing the subalgebras through $t(E_1 \oplus 0_{2,2})$ and $t(0_{2,2} \oplus E_2)$ for equally spaced values of $t \in [\![0, 9]\!]$ and passing it to the matrix exponential which yields invertible matrices of the form $R_{1,t} = e^{tE_1} \oplus I_2$ and $R_{2,t} = I_2 \oplus e^{tE_2}$. We encode an arbitrary initial observation to obtain its representation vector $z$, and traverse the latent space through $R_{i,t}z$. We decode the obtained vectors to obtain the predicted images.

The group algebra offers a smooth parametrization of the group and consequently of the data manifold and enables the prediction of observations evolution in the absence of performed actions. Indeed, in the above example, all transformations can be obtained in the form $\exp(t_1 E_1) \oplus \exp(t_2 E_2)$ for $t_1, t_2 \in \mathbb{R}$.

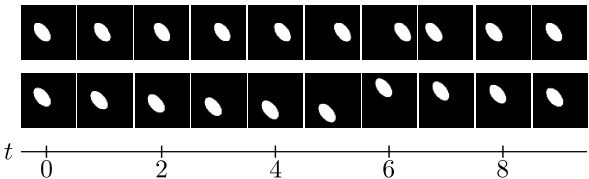

Figure 7: We visualize the linear traversal of the group algebra and its effect on the predicted image reconstruction. The first line corresponds to a traversal $tE_1 \oplus 0_{2,2}$, while the second line corresponds to the traversal $0_{2,2} \oplus tE_2$.

### C.2.4 Additional Experiment

We consider a subset of the dataset consisting of all variations of the heart under a fixed scale. As such the heart is acted on by the group $G = G_\theta \times G_x \times G_y = C_{39} \times C_{32} \times C_{32}$. We train a similar model to the one described in the section C.2.1 by changing the latent space to 6 dimensions, the group representation is fixed of the form $\rho = \rho_1 \oplus \rho_2 \oplus \rho_3$

each of dimension 2. We expect a disentangled representation space $Z = Z_1 \oplus Z_2 \oplus Z_3$. For the visualization of the learned representation manifold Figure 8, we visualize each subspace $Z_i$ separately by only varying one generative factor and keeping all else fixed. We also visualized the learned representations for a subset of transitions corresponding to the elementary generative transitions for each subgroup in Figure 9.

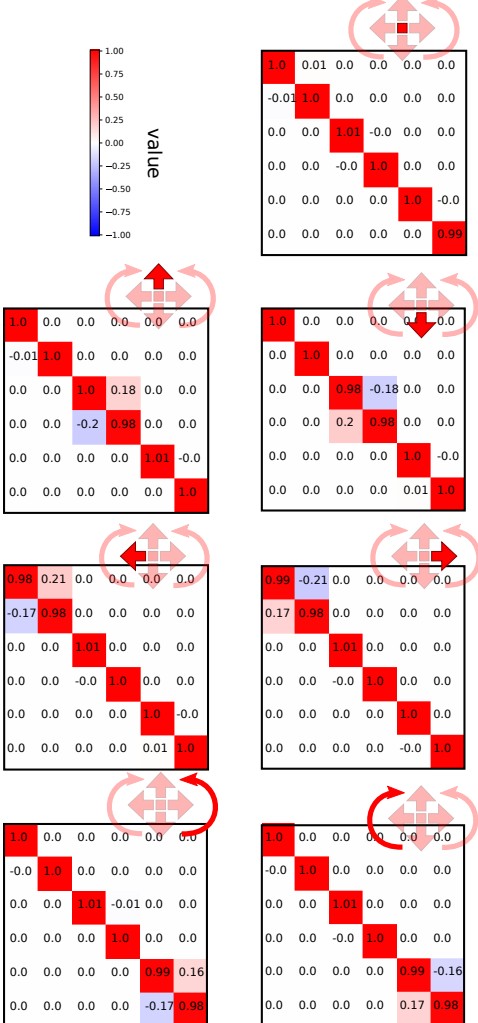

Figure 9: Evaluation of the learned group representation for the identity (upper left) and generative transitions of each subgroup yields disentangled block rotation matrices.

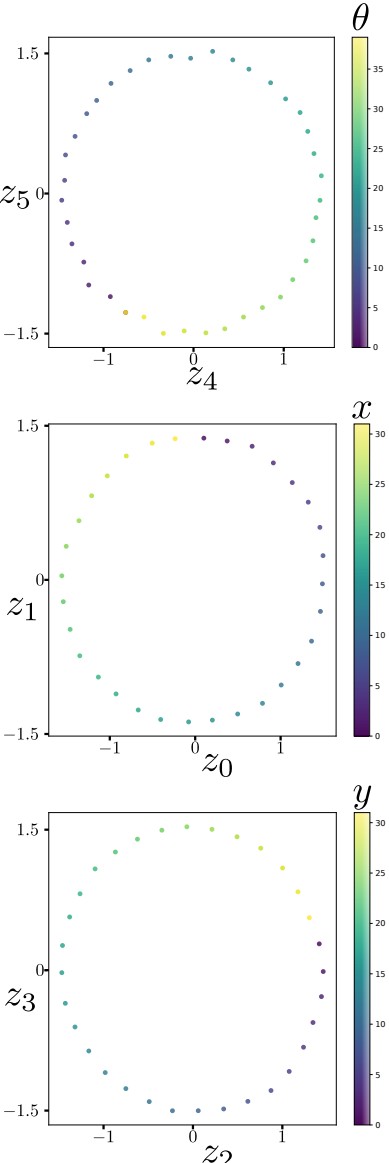

Figure 8: Visualizaion of the 6D embedding vectors for the heart dataset. For each visualized 2D subspace, we only vary the latent represented by the subspace. Color indicates the true latent value.

## C.3 ROLLOUTS PREDICTION

We perform the rollout experiment using the sampling strategy described in Section C.1 where actions were sampled around the identity uniformly. For simplicity, we reduce the range of the actions from $[\![-10, 10]\!]$ to $[\![-3, 3]\!]$. Similarly, we only consider x-y translations in this experiment.

We train each method with a set of pre-generated set of 2-step trajectories and evaluate on a hold-out pre-generated set of 128-step trajectories. For each trajectory, we begin by sampling a random initial state (x,y position) from all possible states.

After that, we sample actions uniformly from the possible actions at each step until the number of steps is satisfied.

For HAE, we map actions to block-diagonal matrices as described in Section 3.2. For the Rotations method [Quessard

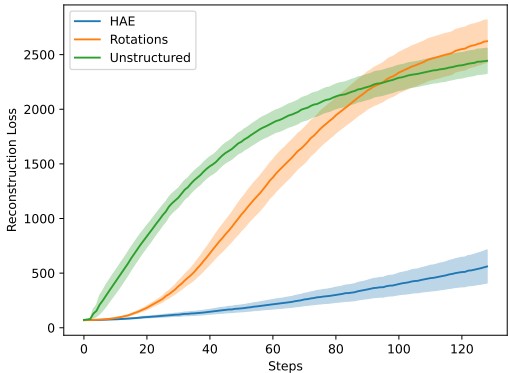

Figure 10: Step-wise reconstruction loss. Lines and shadings represent the mean and one standard error over 15 seeds.

et al., 2020], we map each action to a matrix $\rho(g)$ through a lookup table for all possible actions. For the Unstructured method, we use a 2 layer MLP of size [128, 128] to model the transition by $z_{t+1} = f_\theta(z_t, g_t)$, where we concatenate the latent vector $z_t$ and the one-hot encoding of action $a_t$.

The reconstruction loss is the same for all three methods, as described in Section 3.3. For HAE, we additionally add the latent prediction loss $\mathcal{L}_{pred}$ as described in Section 3.3. We increase $\gamma$ to 1600 which we found to be more stable when matrices are directly parameterized instead of mapped from MLPs. For the Rotations method, an additional entanglement loss $\mathcal{L}_{ent}$ is required to encourage each matrix to act on a specific subspace of the latent space, which is equal to

$$\mathcal{L}_{ent} = \sum_g \sum_{(i,j) \neq (\alpha,\beta)} |\theta_{i,j}^g|^2 \quad \text{with} \quad \theta_{\alpha,\beta}^g = \max_{i,j} |\theta_{i,j}^g|.$$

For the Unstructured method, we only use the reconstruction loss and no additional terms.

We also perform another experiment which adapts the setting of multi-step prediction as in Quessard et al. [2020], where agents can perform multiple simple actions (actions only involve changes in a single generating factor) to the object. In our experiment, we allow the agent to control the object in the dSprite dataset with 7 actions. Namely, translation in the x-y axes, rotation in both directions (clockwise, counter-clockwise), and idle. Each action corresponds to an increment/decrement in one of the generating factors of the dataset, except for idle, which does nothing. Additionally, we use the heart shape from the dataset to fully utilize the orientation latent factor. Figure 10 shows that the Rotations method performs better than the Unstructured method in this setting. This is likely because the actions sampling process satisfies the disentanglement assumption described in Quessard et al. [2020]. However, we see that HAE still outperforms both significantly, suggesting that HAE can also learn efficiently under this setting.

## C.4   MULTI-OBJECTS

We use a similar dataset to the one described in section C.2 except that we use all three shapes: Heart, Ellipse and Square. Because we are considering the action of the same group, observation sequences that start with a given shape have the same shape throughout at different positions. We use a model with a 5D latent, where the last latent is acted on trivially, meaning that the group representation keeps it unchanged. This is equivalent to having a representation of the form $\rho = \rho_x \oplus \rho_y \oplus 1$. We also project the 5D encodings of the dataset on a 2D space through Random Matrix Projection. Note that although the three different tori appear of different sizes, it is only an artifact of the projection while the tori are identical.

## C.5   COMPUTATIONAL RESOURCES

The experiments were performed on an NVIDIA GeForce RTX 3090 and A100 GPUs. The training of our optimal models run for approximately 20 mins.

## D   THIRD-PARTY SOFTWARE

### D.1   DEEP LEARNING FRAMEWORK

To implement our architecture we used the deep learning framework PyTorch. Paszke et al. [2019]

### D.2   HYPERPARAMETER SEARCH

We used the hyperparameter search utility provided in the hypnettorch project https://github.com/chrhenning/hypnettorch/tree/master/hypnettorch/hpsearch to perform a random grid search.

### D.3   DATASET

In the presented experiments, we used the dSprites dataset Matthey et al. [2017]. The dSprites dataset is an image dataset of white sprites on a black background, varying in shape (heart, ellipse, square), in scale (6 values), in orientation (39 values, cyclic), in $x$ and $y$ position (32 values each). We consider all factors besides shape to be cyclic, in particular for the $x$ and $y$ positions, we "glued" opposite borders of images into a torus. The resolution of the images is $64 \times 64$ pixels.

## E   SOCIETAL IMPACT

This work proposes new findings in basic research. To the best of our knowledge, this work does not have immediate

applications with a negative societal impact.