# OpenReview forum: "Homomorphism Autoencoder --- Learning Group Structured Representations from Interactions"
_auai.org/UAI/2022/Workshop/CRL — CRL@UAI 2022 Poster_

### Official Review · Reviewer_1xn7 · 2022-06-25
**Methodological part and mathematical results seem to be coherent**

**Rating:** 6
**Confidence:** 2

**Review:**

The paper is generally clear. The methodological part and the mathematical results seem to be coherent, though I am not familiar with the details of group theory to verify them and go through the proofs. The assumptions and limitations are clearly stated. The overall idea makes sense to me which seems rigorous. There are no problems with experimental part.

I would suggest the authors to explain more about their motivation and contributions in relation with existing works. There are several explanations scattered throughout the paper, e.g. in Section 3, and it would be better to provide a detailed explanation early in Section 1.

---

### Official Review · Reviewer_Gmkq · 2022-06-26
**Review: Homomorphism Autoencoder**

**Rating:** 7
**Confidence:** 4

**Review:**

## Summary
The authors propose a new model, called the Homomorphism Autoencoder, which simultaneously learns a feature extractor (encoder/decoder) and a parameterized representation of how input transformations act on the latent space. They demonstrate that this framework successfully learns a representation which is equivariant with respect to the observed input transformations, can more faithfully produce future rollouts than comparable counterparts, and can learn a symmetry-disentangled representation when multiple object 'identities' are presented during training.

### Originality:
The method proposed in this paper is to the best of my knowledge original as it is explicitly formulated. However, much related work has been done with the same goal (to learn symmetry-based disentangled representations in an unsupervised manner), and using Lie group / Lie algebras, but the paper lacks a significant discussion of related work. This lack of discussion makes it challenging to validate the originality of the work conclusively.

### Clarity:
The paper is very clear and well written. There are a few rough spots that I believe the authors can smooth out with a few proof-reads, and may simply be due to the reduced page limit of the workshop. Some suggestions:
- Page 1, paragraph 2: “The concept of group” → ‘The concept of a group’
- Page 2, paragraph 2: “Au- toencoder”
- Page 7 references: Painter et al. reference is incomplete.

### Significance:
The authors provide a method by which the representation of a group action in latent space can directly be learned. Much prior work has approached this problem from the direction of using a fixed representation of transformations in the latent space, and pushing the problem of matching this into the encoder, while other works have attempted to learn structured transformations such as permutation matrices, which has proved challenging. This work provides a different direction with which to approach this problem and is thus valuable to the community.

### Strengths:
- The paper is well written and has a strong underlying motivation/vision which is clear in the paper’s presentation.
- The authors provide a welcome theoretical analysis of their proposed losses and the implications when they are taken to the limit, furthermore their presentation of this analysis makes sense.
- The empirical results are strong on simple tasks, well presented, and convincing.

### Limitations:
- Code is not provided.
- The related work section of this paper is very weak and should certainly be improved if this is to be submitted as a full conference paper. Although the method is novel to the best of my knowledge, the overlap with prior work is significant and the community would benefit from a brief discussion of the similarity/difference of this method to prior work. Below are just a few (but certainly not all) of the works which I believe are heavily related and should be referenced:
   - Connor et al 2021, Variational Autoencoder with Learned Latent Structure, https://arxiv.org/abs/2006.10597
   - Chau et al. 2020, Disentangling Images with Lie Group Transformations and Sparse Coding, https://arxiv.org/abs/2012.12071
   - Cohen & Welling, 2014, Learning the Irreducible Representations of Commutative Lie Groups. https://arxiv.org/abs/1402.4437
   - Dehmamy et al. 2021, Automatic Symmetry Discovery with Lie Algebra Convolutional Network. https://arxiv.org/abs/2109.07103

- The representation of the group element which the agent observes is a relatively simple transformation of the true element compared with other possible deterministic mappings $\varphi$. A more complex transformation would be a very welcomed experiment to validate to what extent input transformations can truly be learned from raw data.

### Minor suggestions:
- The discussion of 'joint angle' and 'joints space' was more confusing than helpful to me personally, especially when none of the experiments are performed in the RL setting or with an agent. Only after reading the full paper did I understand the reasoning. I would suggest calling this space by a different name and just using the joint analogy as an example.

### Questions:
- In the appendix, you write $\gamma=1600$, is this to simply equate the scales of the two losses? Or is this strong of regularization truly necessary to prevent the encoder from learning a constant representation?

### Conclusion:
The paper is an interesting idea, well executed and presented, and would make a good addition to the workshop – I recommend accept.

---

### Official Review · Reviewer_WBoY · 2022-06-29

**Rating:** 7
**Confidence:** 2

**Review:**

Summary:

This is an interesting paper that deserves to be discussed at the workshop.

The paper develops a method to learn representations based on group theory based definitions of disentanglement. They propose the "Homomorphism Autoencoder" (HAE) which jointly learns (rho, h) where:
- rho is the group representation action which acts on factors z_t to z_{t+1}.
- h is the encoder that extracts a representation from the data x.

rho and h are learned with a two-step objective which combines
1. the latent prediction loss, where rho predicts z_{t+1} from z_t
2. the reconstruction loss

When the group disentanglement definition holds, the group representation action is block diagonal. Inspired by this, the authors enforce rho to be block diagonal. While they do not prove that this block diagonal constraint leads to disentanglement, they show through experiments that "HAE learns a symmetry-based representation (rho , h) that is disentangled with regard to the true factors."

Pros:
- The paper operationalizes group disentanglement ideas in a practical way.  For me, reading this paper brought to light the connections between group disentanglement definitions and mechanism based ideas around identifiability and disentanglement (see e.g. "Properties from mechanisms: an equivariance perspective on identifiable representation learning" (Ahuja, Hartford and Bengio 2021)). It would be interesting to articulate these connections further in future research.
- The paper explains the intuition behind the group representations and transitions well e.g. in Figure 2.

Cons:
- It was not clear to me how exactly rho was estimated.
- It was not clear how the decoder d was estimated (and it wasn't included in the objective optimization?)
- It was not clear to me how the HAE requires the dataset to be organized. It seems that the data needs to be ordered in a specific way (according to the group transitions acting on the factors). Can the authors comment more on this?
- I found it difficult to understand Figures 3 and 6.

---

### Meta-Review · Program_Chairs · 2022-07-06

**Recommendation:** Accept (Poster)
**Confidence:** 4

**Metareview:**

The reviewers agree in judging the work significant and the contribution clear. Empirical results were judged well-presented and convincing. The authors are encouraged to address the reviewers' comments and make the code for their experiments publicly available for the camera-ready version.

---

### Decision · Program_Chairs · 2022-07-06

Accept (Poster)